# HOLOGRAPHIC AND OTHER POINT SET DISTANCES FOR MACHINE LEARNING

## ABSTRACT

We introduce an analytic distance function for moderately sized point sets of known cardinality that is shown to have very desirable properties, both as a loss function as well as a regularizer for machine learning applications. We compare our novel construction to other point set distance functions and show proof of concept experiments for training neural networks end-to-end on point set prediction tasks such as object detection.

## 1 INTRODUCTION

Parametric machine learning models, like artificial neural networks, are routinely trained by *empirical risk minimization*. If we aim to predict an output $y \in \mathcal{Y}$ from an input $x \in \mathcal{X}$, we collect a training set $\mathcal{D}$ of $(x, y)$ pairs and train a parametrized prediction function $h_\theta \colon \mathcal{X} \to \mathcal{Y}$ by minimizing

$$R(\theta) := \frac{1}{|\mathcal{D}|} \sum_{(x,y) \in \mathcal{D}} L(h_\theta(x), y). \tag{1}$$

Here, $L \colon \mathcal{Y} \times \mathcal{Y} \to \mathbb{R}$ is a *loss function* that assigns a scalar loss value to the prediction $\hat{y} = h_\theta(x)$ for the true value $y$. Classical machine learning problems allow for standard choices for the loss function. E.g., for regression problems, where $y, \hat{y} \in \mathbb{R}$, it is common to use the squared loss $L(\hat{y}, y) = (\hat{y} - y)^2$.

Assume, however, that we want to train a machine learning system which—given some input—predicts a *set* of points. In particular, the ordering of the points is neither semantically meaningful nor in any way consistent across data instances. As an example, one may consider an object detection task such as finding the positions of the black stones on an image of a real-world game of Nine Men's Morris. *What should the loss function be in such a case?*

Naively, we might just impose an arbitrary ordering to the sets and treat them as ordered tuples. However, this conceals an important property of the task, the permutation invariance, from our machine learning system. This property might in fact be crucial to learn such a task efficiently. Instead, we would like our loss function to define a meaningful distance between two *sets*, which requires such a loss to be permutation-invariant.

### 1.1 PERMUTATION-INVARIANT LOSS FUNCTIONS

We consider distance functions a for pair of finite sets of points from $\mathbb{R}^d$ and assume the cardinality $N$ to be the same for both sets and known in advance. While we are concerned with unordered sets of points, representing them in a machine-learning context will require us to impose *some* ordering to its elements. Hence, we will represent a set $\{z_1, \ldots, z_N\} \subset \mathbb{R}^d$ of points as an ordered tuple $Z = (z_1, \ldots, z_N)$ and define the shorthand $\mathcal{Z}_N := \left(\mathbb{R}^d\right)^N$ for the space of such tuples.

If a loss function $L(Z, \hat{Z})$ operating on such ordered tuples is supposed to define a meaningful distance between the underlying sets, it will have to be invariant to their arbitrary ordering. Hence, for $Z = (z_1, \ldots, z_N) \in \mathcal{Z}_N$ and a permutation $\sigma \in S_N$, we define $\sigma(Z) := (z_{\sigma(1)}, \ldots, z_{\sigma(N)})$ and demand the following property.

**Definition 1** (Permutation-invariant loss). *We call a function $L : \mathcal{Z}_N \times \mathcal{Z}_N \to \mathbb{R}$ permutation-invariant if and only if for any $\hat{Z}, Z \in \mathcal{Z}_N$ and permutations $\pi, \sigma \in S_N$, we have*

$$L(\pi(\hat{Z}), \sigma(Z)) = L(\hat{Z}, Z). \tag{2}$$

## 1.2 RELATED WORK

While distance measures for sets have been studied in Mathematics and theoretical computer science, they have rarely been investigated in the context of machine learning. A natural way to define a point set distance is to explicitly find a "best" matching between target and prediction points and take the distance between the re-ordered tuples, e.g.,

$$L^{\text{Hung}}(\hat{Z}, Z) = \min_{\sigma \in S_N} \sum_{i \in [N]} \|z_i - \hat{z}_{\sigma(i)}\|^2. \tag{3}$$

This has been proposed under the name *Hungarian loss* by Stewart et al. (2016) for object detection. The name refers to the *Hungarian algorithm* (Kuhn, 1955), which can be used to solve the assignment problem. To our knowledge, that is the first work to propose a permutation-invariant loss function for training a machine learning model. This loss function is intuitive and is a simple, well-conditioned quadratic function of $\hat{Z}$ in regions where the optimal permutation is constant. But it also has its drawbacks. Its gradient is undefined at "transition points" with more than one optimal permutation. More importantly, having to solve the assignment problem (Eq. 3) for each single evaluation of the loss function is problematic in a machine learning context, where we nowadays rely on simple, highly-efficient computational modules in frameworks such as TensorFlow (Abadi et al., 2016). It can also be computationally slow, since the Hungarian algorithm is $\mathcal{O}(N^3)$ and does not lend itself to a highly-parallelized implementation on GPUs.

Rezatofighi et al. (2018) propose to use a neural network to jointly model the elements of a set as well as the permutation in which they appear. This approach gives rise to an alternating optimization procedure, where the first step solves for an optimal permutation using the Hungarian algorithm and the second step updates the weights of the neural network. We note that their probabilistic formulation provides an elegant framework to learn the cardinality of the set alongside its elements. While this is certainly important in practice, in this paper we focus on investigating properties of some point set distance functions, assuming the cardinality to be known. Future work could integrate these loss functions with the cardinality learning procedure of Rezatofighi et al. (2018) to replace the cumbersome inner-loop solution of an optimal assignment problem.

Zaheer et al. (2017) discuss neural network architectures that take sets as *inputs*. They find a general characterization of permutation-invariant functions and use that insight to devise neural network architectures which operate on sets in a permutation-invariant fashion.

## 1.3 SCOPE OF THIS WORK

In this work, we consider some alternative permutation-invariant point set distance functions, all of which have running time $\mathcal{O}(N^2)$ and can be implemented with operations that are readily available in automatic differentiation frameworks like TensorFlow. In particular, they do not rely on the solution of an assignment problem akin to Eq. (3). We pay specific attention to properties relevant to machine learning, in particular, how amenable such distances are to gradient-based numerical optimization.

As one option, we propose a novel distance function for point sets, which we name "holographic loss", since it is a metric distance on fingerprints of point sets that "holographically" encodes their structure, i.e. moving any point in the set will collectively (and analytically) change all entries in the fingerprint-vector. We show that it has favorable properties in addition to being analytic everywhere, such as having a diagonal Hessian at the minima, which furthermore all are global and correspond to exactly matched up point sets. Also, this loss function has a natural generalization to multi-sets.

We present proof of concept experiments on a simple object detection task based on MNIST digits, where we train a convolutional neural network to directly predict the locations of digits in an image. These experiments show that end-to-end training with a simple permutation-invariant loss function is a viable approach for problems that can be formulated as a point set prediction task.

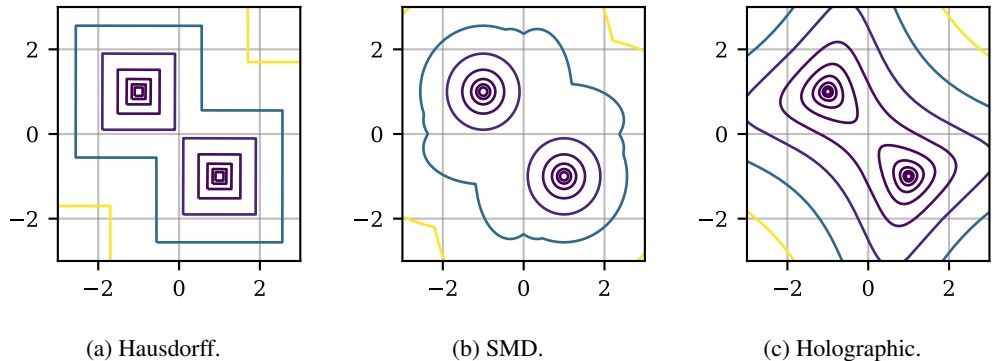

<table>
<tr><td>(a) Hausdorff.</td><td>(b) SMD.</td><td>(c) Holographic.</td></tr>
</table>

Figure 1: Contour plots of different permutation-invariant point set loss function on a simple problem of matching two one-dimensional target points located at $-1$ and $1$.

## 2 HAUSDORFF DISTANCE AND SUM OF MINIMAL DISTANCES-SQUARED

Distance functions for sets have been studied at least since the days of Hausdorff's 1914 book on set theory Hausdorff (1914). Considering Hausdorff's construction in our context, i.e. as a (2nd order) loss function for finite point sets, the one-sided Hausdorff distance is given by $L^{\text{OSH}}(\hat{Z}, Z)^2 = \max_{i \in [N]} \min_{j \in [N]} \|z_i - \hat{z}_j\|^2$, which is the maximum squared distance of any target point to its closest prediction point. The symmetric Hausdorff distance is

$$L^{\text{H}}(\hat{Z}, Z)^2 = \max\left(L^{\text{OSH}}(\hat{Z}, Z)^2, L^{\text{OSH}}(Z, \hat{Z})^2\right). \tag{4}$$

$L^{\text{H},2}$ is easily computed with $\mathcal{O}(N^2)$ effort, almost everywhere differentiable, and makes intuitive sense as a machine learning loss function. However, it has some properties that might not be desirable in a machine learning context. Since the Hausdorff distance is determined by a single "worst-offender" pair of target and prediction points, each gradient descent step will only adjust the position of a single prediction point, until there is an almost-tie between different point-pairs, and henceforth subsequent gradient steps will jump around and keep almost-tied point pairs almost-tied. It stands to reason that this might, especially with some optimization strategies, be undesirable compared to loss functions whose gradient collectively moves all points.

A possible remedy to that problem is to sum the minimal distances-squared instead of taking their maximum. The one-sided sum of minimal distances-squared is $L^{\text{OSSMD}}(\hat{Z}, Z)^2 = \frac{1}{N} \sum_{i=1}^{N} \min_{j \in [N]} \|z_i - \hat{z}_j\|^2$ and the (symmetric) sum of minimal distances-squared is

$$L^{\text{SMD}}(\hat{Z}, Z)^2 = \frac{1}{2}\left(L^{\text{OSSMD}}(\hat{Z}, Z)^2 + L^{\text{OSSMD}}(Z, \hat{Z})^2\right). \tag{5}$$

We do not consider the one-sided variants of these two distance functions in our comparison. The one-sided variants centered on the prediction points will not make sense as a machine learning loss function, since they become zero when matching all predicted points to any one of the target points, e.g., $\hat{z}_j = z_1$ for all $j$. The one-sided variants centered on the target points, while more sensible, have simple failure cases as well, such as the one depicted in Figure 4. We thus restrict our attention to the symmetric loss functions $L^{\text{H}}$ and $L^{\text{SMD}}$. Figures 1a and 1b show contour plots for these two functions on a simple problem with two one-dimensional target points, while 1c shows our construction, described in the next section.

## 3 "HOLOGRAPHIC" LOSS

### 3.1 SET FINGERPRINTS

A general strategy to devise permutation-invariant loss functions is to (differentiably) map tuples $Z$ representing a point set (or multi-set) to a "fingerprint" vector $f(Z)$ in some fingerprint-space $\mathbb{F}$

such that $f(\hat{Z}) = f(Z)$ if and only if $\hat{Z} = \sigma(Z)$ for some permutation $\sigma$. A permutation-invariant loss function can then be defined via (some) metric distance in $\mathbb{F}$. Any such loss function will inherit properties of the distance in $\mathbb{F}$, such as positive definiteness, symmetry, or the triangle inequality.

In one dimension, a suggestive choice for a fingerprint function on tuples of $N$ points would be a vector of moments $M_k(Z) := \sum_{i=1}^{N} z_i^k$. The mathematical appendix of Zaheer et al. (2017) identifies the moments of points in a set as an universal permutation-invariant fingerprint, in the sense that every other such fingerprint can be defined in terms of these quantities. This suggest using squared-euclidean-distance-of-moments-vectors as a loss function. However, this has some fundamental problems, such as poor conditioning, and it is non-trivial to generalize to higher-dimensional points. See Appendix B for a detailed discussion.

## 3.2 EXPLOITING THE 2-D {POINT SET} $\Leftrightarrow$ {POLYNOMIAL} CORRESPONDENCE

In this section, we want to focus our attention to two-dimensional point sets (or multi-sets), and consider one-dimensional sets as a special case with all points having a 2nd coordinate of zero. We will subsequently consider generalizations to higher dimensions.

We start by identifying a two-dimensional point $z = (x, y) \in \mathbb{R}^2$ with the complex number $z = x + iy \in \mathbb{C}$. With this, we can map any tuple $Z = (z_1, \ldots, z_N)$ that represents a (multi-)set to a complex polynomial

$$P_Z(u) = (u - z_1)(u - z_2) \cdots (u - z_N). \tag{6}$$

Note that $P_Z = P_{\hat{Z}}$ if and only if $\hat{Z} = \sigma(Z)$ for some permutation $\sigma \in S_N$. Furthermore, such a monic (i.e. having leading coefficient 1) polynomial of degree $N$ is uniquely determined by its values at $N$ distinct points. This suggests to choose a set $U = (u_1, \ldots, u_N)$ of $N$ distinct evaluation points and to define a fingerprint as

$$f(Z) = (P_Z(u_1), \ldots, P_Z(u_N)). \tag{7}$$

Taking the real-valued $L_2$ distance in $\mathbb{C}^N$ gives us the "holographic" loss function

$$L_U^{\text{Hol}}(\hat{Z}, Z)^2 = \frac{1}{N} \sum_{i=1}^{N} |P_Z(u_i) - P_{\hat{Z}}(u_i)|^2 = \frac{1}{N} \sum_{i=1}^{N} \left( P_Z(u_i) - P_{\hat{Z}}(u_i) \right)^* \cdot \left( P_Z(u_i) - P_{\hat{Z}}(u_i) \right). \tag{8}$$

While the underlying fingerprint is holomorphic, i.e. complex differentiable, the distance-squared function is not, due to anti-linearity of the underlying complex scalar product in the first argument. The adjective 'holographic' here refers to this function depending collectively on the locations of all points, unlike Hausdorff distance, which for a generic configuration will not change when slightly changing the positions of any points other than the specific pair that determines its value. Also, for $U \neq Z$, correctly matching up the candidate and target points means matching up both the complex amplitude as well as the complex phase at the evaluation-points. However, the analogy with holography ends here, there is no deeper correspondence beyond that.

**Proposition 1.** *The Holographic loss trivially has the following properties:*

- *Positive definiteness: $L_U^{Hol}(\hat{Z}, Z)^2 \geq 0$ and $L_U^{Hol}(\hat{Z}, Z)^2 = 0$ if and only if $\hat{Z} = \sigma(Z)$.*

- *Symmetry: $L_U^{Hol}(\hat{Z}, Z)^2 = L_U^{Hol}(Z, \hat{Z})^2$*

- *Triangle inequality: $\sqrt{L_U^{Hol}(\hat{Z}, Z)^2} \leq \sqrt{L_U^{Hol}(\hat{Z}, Z')^2} + \sqrt{L_U^{Hol}(Z', Z)^2}.$*

- *$L_U^{Hol}(\hat{Z}, Z)^2$ is quadratic in each individual $\hat{z}_i$'s real and also imaginary part when keeping all other point-coordinates fixed.*

## 3.3 EVALUATING AT THE TARGET POINTS

If the target points form a set rather than a multi-set (i.e. there are no duplicates), we have the option to choose the set $U$ of evaluation points as identical to the target points $Z$. Then, since $P_Z(z_i) = 0$

by construction, we get the simpler loss function

$$L^{\text{Hol}}(\hat{Z}, Z)^2 = \frac{1}{N} \sum_{i=1}^{N} |P_{\hat{Z}}(z_i)|^2 = \frac{1}{N} \sum_{i=1}^{N} \left| \prod_{j=1}^{N} (z_i - \hat{z}_j) \right|^2 = \frac{1}{N} \sum_{i=1}^{N} \prod_{j=1}^{N} |z_i - \hat{z}_j|^2. \quad (9)$$

In the two-dimensional case, note that the squared absolute value of the complex number $z_i - \hat{z}_j$ is identical to the squared Euclidean distance $\|z_i - \hat{z}_j\|^2$ of the two points in $\mathbb{R}^2$:

$$L^{\text{Hol}}(\hat{Z}, Z)^2 = \frac{1}{N} \sum_{i=1}^{N} \prod_{j=1}^{N} \|z_i - \hat{z}_j\|^2. \quad (10)$$

This *target-centered* version will be used in the practical applications presented here. For each target point, we take the product of squared distances to all prediction points and then sum this quantity over all target points. If a target point $z_i$ is matched closely by *any* of the prediction points, the product of squared distances will tend to be small and, consequently, $z_i$ will not contribute significantly to the loss. Note that, unlike Eq. (8), this loss function will no longer be symmetric. Other than that, it inherits the properties of Proposition 1.

### 3.4 HIGHER-DIMENSIONAL POINTS

There is no higher-dimensional equivalent of the polynomial construction in Eq. (6), since there is no known generalization of the Fundamental theorem of Algebra to higher-dimensional real algebras beyond $D = 2$. However, we can heuristically extend Eq. (10), which is a simple sum of products of squared Euclidean distances, to point sets of arbitrary dimension. As we show in the next section, for $D = 2$, the correspondence between point sets and polynomials can be used to prove some favorable properties of the loss function. We do not yet have a proof for these properties for $D \neq 2$.

### 3.5 GRADIENTS AND STATIONARY POINTS

The gradient of the Holographic loss with an arbitrary set of evaluation points (Eq. 8) can be computed using complex backpropagation of errors, see Appendix A. Here, we restrict our attention to the gradient of Eq. (10) with respect to $\hat{x}_k$ and $\hat{y}_k$, which evaluates to

$$\left( \frac{\partial L^{\text{Hol}}(\hat{Z}, Z)^2}{\partial \hat{x}_k}, \frac{\partial L^{\text{Hol}}(\hat{Z}, Z)^2}{\partial \hat{y}_k} \right) = -\frac{2}{N} \sum_{i=1}^{N} \left( \prod_{j=1, j \neq k}^{N} \|z_i - \hat{z}_j\|^2 \right) (x_i - \hat{x}_k, y_i - \hat{y}_k). \quad (11)$$

This gradient has a rather intuitive interpretation (also for $D > 2$). The negative gradient is a weighted sum of the vectors $(z_i - \hat{z}_k)$, which pulls the prediction $\hat{z}_k$ in the direction of target point $z_i$. The weight of this attractive force in the direction of $z_i$ is $\propto \prod_{j \neq k} \|z_i - \hat{z}_j\|^2$, which measures how well point $z_i$ is already matched by any prediction point *other* than $\hat{z}_k$. Unlike the Hausdorff distance, the gradient of the Holographic loss thus adjusts the position of all points simultaneously. We can get some intuition for the dynamics induced by this gradient from example solutions of the gradient flow ODE, see Appendix C.

**Stationary Points**  Obviously, any permutation-invariant loss function will be non-convex by the mere fact that it has multiple global minima. Beyond that, the Holographic loss will have additional stationary points. For example, for $Z = ((0,0), (1,1))$, the gradient vanishes at the non-optimal configuration $\hat{Z} = ((0.5, 0.5), (0.5, 0.5))$. The following Proposition shows that, for $D = 2$, these non-optimal stationary points can *not* be local minima. The proof (see Appendix A) relies on complex polynomials and, thus, only holds for two-dimensional point sets. While we conjecture that this property holds for the Holographic loss in Eq. (10) for points of any dimension, we currently do not have a proof for this.

**Proposition 2.** *Let $D = 2$ and point sets $Z, U$ be given, possibly $U = Z$. Then $Z^*$ is a local minimum of the function $\hat{Z} \mapsto L_U^{Hol}(\hat{Z}, Z)^2$ if and only if $Z^* = \sigma(Z)$ for some permutation $\sigma \in S_N$. In this case we have $L_U^{Hol}(Z^*, Z)^2 = 0$ and, thus, all minima are global.*

Table 1: Properties of point set loss functions.

| Loss Function | Complex. | Analytic | GPU-Friendly | Non-Sparse Grad. | Multi-Set Compatible |
|---|---|---|---|---|---|
| **Hungarian** | $\mathcal{O}(N^3)$ | No | No | Yes | Yes |
| **Hausdorff** | $\mathcal{O}(N^2)$ | No | Yes | No | No |
| **SMD** | $\mathcal{O}(N^2)$ | No | Yes | Yes | No |
| **Holographic** | $\mathcal{O}(N^2)$ | Yes | Yes | Yes | in $D = 2$ |

### 3.6 BEHAVIOR NEAR MINIMA

The following proposition characterizes the behavior of the Holographic loss near optima up to second order. The proof can be found in Appendix A.

**Proposition 3.** *Let $Z$ be fixed and define $L_Z(\hat{Z})^2 = L^{Hol}(\hat{Z}, Z)^2$, where $\hat{Z}$ is treated as a $\mathbb{R}^{D \cdot N}$ vector. At any global optimum where $\hat{Z} = \sigma(Z)$ with $\sigma \in S_N$, the Hessian matrix $\partial_j \partial_k L_Z(\hat{Z})^2 \in \mathbb{R}^{D \cdot N \times D \cdot N}$ is diagonal and its elements are*

$$\frac{2}{N} \prod_{j \in [N] \setminus \{i\}} \|z_i - z_j\|^2, \quad i \in [N], \tag{12}$$

*each appearing $D$ times.*

The nature of the diagonal elements, which are also the eigenvalues of the Hessian, suggests that the problem can become ill-conditioned if the pairwise distances between points have vastly different scales, e.g., if there is a pair of points that is very close to each other relative to all the other points. One might even be worried whether minimizing the Holographic loss to numerical accuracy correctly matches up point sets in such pathological situations. In Appendix B, we show that this not a problem even for sets with about $40$ randomly sampled points. This behavior is to a large extent due to the property of the Hessian always being diagonal at minima. Since the eigendirections of the Hessian coincide with the coordinate directions, floating point numerics can represent the gradients in these eigendirections with high relative precision despite their vastly different scales.

This concludes our description of the Holographic loss. Table 1 gives an overview of different properties of the loss functions under consideration.

## 4 EXPERIMENTS

We apply the three point set loss functions—Hausdorff, sum of minimal distances-squared, and Holographic loss—to a simple object detection toy tasks. The purpose of these experiments is to demonstrate the viablity of end-to-end training with simple permutation-invariant loss functions without explicitly modelling permutations or having to solve optimal assignment problems.

Object detection is one of the most prominent tasks in the field of computer vision. Given an input image, the task is to predict the locations (e.g., center points or bounding boxes) of objects shown in this image. Since there usually will not be any meaningful or consistent ordering of objects in an image, this is inherently a point set prediction task. However, existing approaches to object detection avoid treating it as such. Instead, state-of-the-art object detectors are carefully engineered systems combining multiple components. R-CNN (Girshick et al., 2014; Girshick, 2015; Ren et al., 2015) generates proposal regions hypothesizing object locations and subsequently assigns a score to each region *independently*, indicating how likely it is to actually contain an object. YOLO (Redmon et al., 2016) divides an image into a grid and asks each grid cell to predict a pre-specified number of likely object bounding boxes, together with confidence scores. For both R-CNN and YOLO, generating a final output set involves post processing steps like non-maximum suppression. None of these systems learn the map from an image input to the object locations end-to-end. We want to demonstrate that this is possible to do with any of the three point set loss functions discussed in this paper.

**Dataset** For this proof of concept experiment, we create a simple object detection data set, MNIST-DETECT, with a fixed number of objects (MNIST digits) in each image. To generate an MNISTDE-TECT image, we sample four MNIST digits, crop them to a tight bounding box and rescale them

Table 2: MNISTDetect test set results. Each line corresponds to the same neural network trained with a specific loss function.

|  | Centers (Hung. loss) | Bounding boxes (detection rate) |
| --- | --- | --- |
| **Trained on Hausdorff** | 7.71 | 76.03 |
| **Trained on SMD** | 14.67 | 80.64 |
| **Trained on Holographic** | 7.89 | 79.20 |

by a factor chosen uniformly at random from $\{0.5, 0.6, 0.7, 0.8, 0.9\}$. We then place them in random locations of a $50 \times 50$ pixels image, allowing overlap of the bounding boxes, but not of the actual digits. Finally, we add noise to the image. Figure 2 shows some examples. Each example is annotated with the center location as well as the bounding box of each digit with no particular ordering.

**Neural Architecture and Training**    We use a very simple convolutional neural network that takes such an MNISTDETECT image as input and applies three convolutional layers (with 64 filters of $3 \times 3$ pixels each) and two fully-connected layers (500 and 200 neurons, respectively), all with ReLU activation. The output layer is of size $4D$ with no non-linearity and is reshaped to a $4 \times D$ array, where each row is supposed to predict the location of one of the four digits in the image. We performed separate experiments for predicting center locations ($D = 2$) and bounding boxes ($D = 4$). The network is trained end-to-end using Hausdorff, SMD, and Holographic loss. For each loss function, we train the network for a fixed number of epochs using the Adam optimizer (Kingma & Ba, 2014) and a fixed mini-batch size of 128. We evaluate the loss on a validation set after each epoch of training and retain the weights that achieve minimal validation loss. The step size is tuned for each loss function independently via a grid search, but turned out to be the same (0.001) for all loss functions.[1]

**Evaluation**    For each loss function, we evaluate the network that achieved minimal validation loss on a held-out test set. As "impartial" comparison metrics, we report the Hungarian loss for center point prediction and detection rate for bounding box prediction. In the latter case, we count a digit as successfully detected if its bounding box has an intersection over union (IoU) larger than 0.5 with any of the predicted bounding boxes.

Table 2 shows quantitative results and Figure 2 depicts some qualitative results for bounding box prediction. With all three loss functions, the neural network manages to learn the task reasonably well even though it is a simplistic CNN architecture that has not been tuned to the task at all.

## 5    CONCLUSION

We discussed a novel point set loss function, which is analytic everywhere, vis-a-vis two other simple alternatives with matching computational complexity for point set prediction tasks as alternatives to more involved approaches (Stewart et al., 2016; Rezatofighi et al., 2018). Proof of concept experiments showed that end-to-end training with such simple loss functions is a viable approach for point set prediction tasks, such as object detection. We expect that simple constructions such as the "holographic" point set distance introduced here may turn out useful not only for point set predictions, but also as a regularizer, for example to encourage (unsupervised) clustering to align its clusters with the clusters found by an earlier version of the model.

## REFERENCES

Martín Abadi, Paul Barham, Jianmin Chen, Zhifeng Chen, Andy Davis, Jeffrey Dean, Matthieu Devin, Sanjay Ghemawat, Geoffrey Irving, Michael Isard, et al. Tensorflow: a system for large-scale machine learning. In *OSDI*, volume 16, pp. 265–283, 2016.

---

[1]This stands to reason, since we used the same architecture with all loss functions and the Adam optimizer is invariant to a rescaling of its objective function.

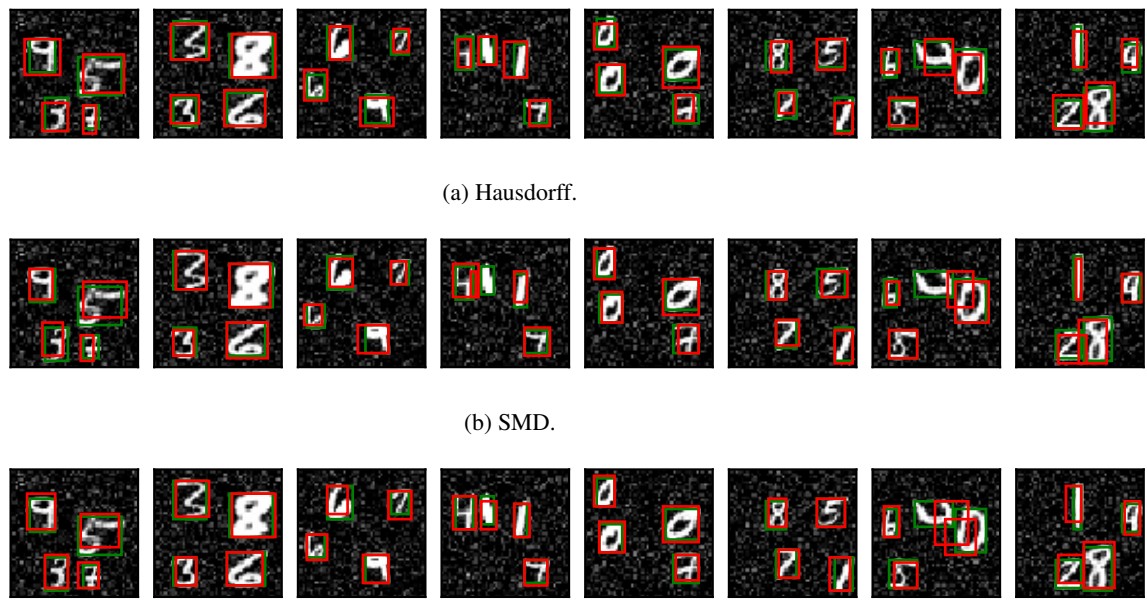

(a) Hausdorff.

(b) SMD.

(c) Holographic.

Figure 2: Qualitative test set results for bounding box prediction on the MNISTDETECT dataset with a convolutional neural network, trained end-to-end with different loss functions. Groundtruth bounding boxes in green, predictions in red.

Vladimir Arnold. On functions of three variables. In *Proceedings of the USSR Academy of Sciences*, volume 114, pp. 679–681, 1957.

Branko Ćurgus and Vania Mascioni. Roots and polynomials as homeomorphic spaces. *Expositiones Mathematicae*, 24(1):81–95, 2006.

Ross Girshick. Fast R-CNN. In *Proceedings of the IEEE International Conference on Computer Vision*, pp. 1440–1448, 2015.

Ross Girshick, Jeff Donahue, Trevor Darrell, and Jitendra Malik. Rich feature hierarchies for accurate object detection and semantic segmentation. In *Proceedings of the IEEE conference on computer vision and pattern recognition*, pp. 580–587, 2014.

Felix Hausdorff. *Grundzüge der Mengenlehre*. 1914.

Raphael Hunger. *An introduction to complex differentials and complex differentiability*. Munich University of Technology, Inst. for Circuit Theory and Signal Processing, 2007.

Diederik P Kingma and Jimmy Ba. Adam: A method for stochastic optimization. *arXiv preprint arXiv:1412.6980*, 2014.

Andrey Kolmogorov. On the representations of continuous functions of several variables by superpositions of continuous functions of fewer variables. In *Proceedings of the USSR Academy of Sciences*, volume 108, pp. 179–182, 1956.

Harold W Kuhn. The Hungarian method for the assignment problem. *Naval research logistics quarterly*, 2(1-2):83–97, 1955.

Joseph Redmon, Santosh Divvala, Ross Girshick, and Ali Farhadi. You only look once: Unified, real-time object detection. In *Proceedings of the IEEE conference on computer vision and pattern recognition*, pp. 779–788, 2016.

Shaoqing Ren, Kaiming He, Ross Girshick, and Jian Sun. Faster R-CNN: Towards real-time object detection with region proposal networks. In *Advances in Neural Information Processing Systems*, pp. 91–99, 2015.

S Hamid Rezatofighi, Roman Kaskman, Farbod T Motlagh, Qinfeng Shi, Daniel Cremers, Laura Leal-Taixé, and Ian Reid. Deep perm-set net: Learn to predict sets with unknown permutation and cardinality using deep neural networks. *arXiv preprint arXiv:1805.00613*, 2018.

Russell Stewart, Mykhaylo Andriluka, and Andrew Y Ng. End-to-end people detection in crowded scenes. In *Proceedings of the IEEE conference on computer vision and pattern recognition*, pp. 2325–2333, 2016.

Manzil Zaheer, Satwik Kottur, Siamak Ravanbakhsh, Barnabas Poczos, Ruslan R Salakhutdinov, and Alexander J Smola. Deep sets. In I. Guyon, U. V. Luxburg, S. Bengio, H. Wallach, R. Fergus, S. Vishwanathan, and R. Garnett (eds.), *Advances in Neural Information Processing Systems 30*, pp. 3391–3401. Curran Associates, Inc., 2017. URL `http://papers.nips.cc/paper/6931-deep-sets.pdf`.

# A   MATHEMATICAL DETAILS

## A.1   PROOF OF PROPOSITION 2

We will need the following Lemma, which states that the roots of a complex polynomial continuously depend on its coefficients.

**Lemma 1.** *Let $F(u) = u^N + \sum_{k=0}^{N-1} c_k u^k$ be a monic complex polynomial and factor it as $F(u) = (u - a_1)(u - a_2) \cdots (u - a_N)$ with $a_k \in \mathbb{C}$ some ordering of the roots. Then, for every $\varepsilon > 0$ there exists $\delta > 0$ such that every polynomial $G(u) = u^N + \sum_{k=0}^{N-1} d_k u^k$ with $|d_k - c_k| < \delta$ can be written as $G(u) = (u - b_1)(u - b_2) \cdots (u - b_N)$ where $|b_k - a_k| < \varepsilon$.*

*Proof.* This is a reformulation of the well-known continuity result, a proof of which can be found, for example, in Ćurgus & Mascioni (2006). One notes that, to first order in a small shift $\epsilon$ in the coefficients of a polynomial, the shift of the zeros can be found by a single step of Newton-Raphson iteration, except at higher-degree zeros (where the derivative of the polynomial in the denominator also has a zero). □

We can now prove the Proposition.

*Proof or Proposition 2.* We abbreviate $L(\hat{Z}, Z)^2 := L_U^{\text{Hol}}(\hat{Z}, Z)^2 = \frac{1}{N} \sum_{i=1}^{N} |P_{\hat{Z}}(u_i) - P_Z(u_i)|^2$. We already observed that $L(\hat{Z}, Z)^2 = 0$ if and only if $\hat{Z} = \sigma(Z)$ for some $\sigma \in S_N$. Now assume $L(\hat{Z}, Z)^2 \neq 0$. To see that $\hat{Z}$ can not be a local minimum, we need to show that for any $\varepsilon > 0$ there is $Z'$ with $d(\hat{Z}, Z') < \varepsilon$ such that $L(Z', Z) < L(\hat{Z}, Z)$, where $d(\hat{Z}, Z') := \sum_{i=1}^{N} |\hat{z}_i - \hat{z}_i'|^2$. To construct such a $Z'$, we define the polynomial $Q_\lambda \colon \mathbb{C} \to \mathbb{C}$,

$$Q_\lambda(u) := P_{\hat{Z}}(u) + \lambda(P_Z(u) - P_{\hat{Z}}(u)), \quad \lambda \in [0, 1], \tag{13}$$

which linearly interpolates between $P_{\hat{Z}}$ and $P_Z$. Note that $Q_\lambda$ is monic for $\lambda \in [0, 1]$ and $Q_0 = P_{\hat{Z}}$. Let $Z'_\lambda$ be some ordering of the roots of $Q_\lambda$. Then

$$L(Z'_\lambda, Z) = \frac{1}{N} \sum_{i=1}^{N} |Q_\lambda(u_i) - P_Z(u_i)|^2 = \frac{1}{N} \sum_{i=1}^{N} |P_{\hat{Z}}(u_i) + \lambda(P_Z(u_i) - P_{\hat{Z}}(u_i)) - P_Z(u_i)|^2$$

$$= (1 - \lambda)^2 \frac{1}{N} \sum_{i=1}^{N} |P_{\hat{Z}}(u_i) - P_Z(u_i)|^2 = (1 - \lambda^2) L(\hat{Z}, Z).$$

$$\tag{14}$$

Since we assumed $L(\hat{Z}, Z) > 0$, this means that $L(Z'_\lambda, Z) < L(\hat{Z}, Z)$ for any $\lambda > 0$. It remains to show that for any $\varepsilon > 0$ there is $\lambda_\varepsilon > 0$ and a permutation $\sigma$ such that $d(\hat{Z}, \sigma(Z'_{\lambda_\varepsilon})) < \varepsilon$. However, since the coefficients of $Q_\lambda$ continuously depend on $\lambda$, this follows as a simple consequence of Lemma 1. □

## A.2   PROOF OF PROPOSITION 3

*Proof of Proposition 3.* Denote by $z_{k,l} := [\hat{z}_k]_l$ the $l$-th coordinate of the $k$-th prediction point (and likewise for target points). We have previously found the first derivatives of the loss w.r.t. the elements of $\hat{Z}$ to be

$$\frac{\partial L_Z(\hat{Z})^2}{\partial \hat{z}_{k,l}} = -\frac{2}{N} \sum_{i=1}^{N} \left( \prod_{j=1, j \neq k}^{N} \|z_i - \hat{z}_j\|^2 \right) (z_{i,l} - \hat{z}_{k,l}). \tag{15}$$

We can easily calculate the second derivatives to be

$$
\frac{\partial^2 L_Z(\hat{Z})^2}{\partial \hat{z}_{k,l}^2} = \frac{2}{N} \sum_{i=1}^{N} \left( \prod_{j=1, j\neq k}^{N} \|z_i - \hat{z}_j\|^2 \right),
$$

$$
\frac{\partial^2 L_Z(\hat{Z})^2}{\partial \hat{z}_{k,m} \partial \hat{z}_{k,l}} = 0, \quad m \neq l, \tag{16}
$$

$$
\frac{\partial^2 L_Z(\hat{Z})^2}{\partial \hat{z}_{k',m} \partial \hat{z}_{k,l}} = \frac{4}{N} \sum_{i=1}^{N} \left( \prod_{j\notin\{k,k'\}} \|z_i - \hat{z}_j\|^2 \right) (z_{i,m} - \hat{z}_{k',m})(z_{i,l} - \hat{z}_{k,l}), \quad k' \neq k.
$$

At any optimum, the latter becomes zero, since for each $z_i$ there is exactly one $j \in [N]$ such that $\hat{z}_j = z_i$. This makes the Hessian matrix diagonal. Furthermore, the diagonal elements simplify at an optimum: Letting $\sigma$ be the specific permutation of this optimum, such that $\hat{z}_j = z_{\sigma(j)}$, we get

$$
\frac{\partial^2 L_Z(\hat{Z})^2}{\partial \hat{z}_{k,l}^2} = \frac{2}{N} \sum_{i=1}^{N} \left( \prod_{j=1, j\neq k}^{N} \|z_i - z_{\sigma(j)}\|^2 \right). \tag{17}
$$

The product becomes zero for all but one $i$: the one where $\sigma(j) \neq i$ for all $j \in [N]\backslash\{k\}$, which is exactly $\sigma^{-1}(k)$. The diagonal element becomes

$$
\frac{\partial^2 L_Z(\hat{Z})^2}{\partial \hat{z}_{k,l}^2} = \frac{2}{N} \prod_{j\neq k} \|z_{\sigma^{-1}(k)} - z_{\sigma(j)}\|^2 = \frac{2}{N} \prod_{j\neq\sigma^{-1}(k)} \|z_{\sigma^{-1}(k)} - z_j\|^2. \tag{18}
$$

This is independent of $l$ and thus appears $D$ times for $l = 1, \ldots, D$ and the set of diagonal elements is the same for each optimum (different optima correspond to different permutations $\sigma$, which only changes the ordering). $\qquad\square$

### A.3 Gradients of the Holographic Loss

In $D = 2$, the Holographic loss (Eq. 8) is polynomial in the $2N$ real coordinates of $N$ points, so a gradient can be obtained via backpropagation in the usual way. When using a set of evaluation-points $U$ that does not coincide with the target points $Z$, there is some additional structure available due to the loss being based on a complex polynomial (hence complex-differentiable) that can be exploited to structurally simplify the computation. To apply *Wirtinger calculus* (a readable introduction can be found in Hunger (2007)), we rewrite our loss function (considering $Z$ and $U$ fixed)

$$
L(Z) := L_U^{\text{Hol}}(\hat{Z}, Z)^2 = \sum_{i=1}^{N} |P_Z(u_i) - P_{\hat{Z}}(u_i)|^2 = \sum_{i=1}^{N} \left| \prod_{j=1}^{N}(u_i - z_j) - \prod_{j=1}^{N}(u_i - \hat{z}_j) \right|^2
$$

$$
= \sum_{i=1}^{N} \left( \left[ \prod_{j=1}^{N}(u_i - z_j) - \prod_{j=1}^{N}(u_i - \hat{z}_j) \right] \left[ \prod_{j=1}^{N}(u_i^* - z_j^*) - \prod_{j=1}^{N}(u_i^* - \hat{z}_j^*) \right] \right) \tag{19}
$$

as $L(\hat{Z})^2 = \tilde{L}(\hat{Z}, \hat{Z}^*)^2$, where

$$
\tilde{L}(V, W)^2 := \sum_{i=1}^{N} \left( \left[ \prod_{j=1}^{N}(u_i - z_j) - \prod_{j=1}^{N}(u_i - v_j) \right] \left[ \prod_{j=1}^{N}(u_i^* - z_j^*) - \prod_{j=1}^{N}(u_i^* - w_j) \right] \right) \tag{20}
$$

Then, $\tilde{L}^2$ is a complex-differentiable function on $\mathbb{C}^N \times \mathbb{C}^N$, and the usual reasoning for sensitivity backpropagation also applies in this complex case.

Informally, denote the sensitivites of $\tilde{L}^2$ w.r.t. $v_j$ and $w_j$ by $\sigma_{v_j}$ and $\sigma_{w_j}$. It turns out that $\sigma_{w_j} = \sigma_{v_j}^*$. Now, we ultimately want to know the sensitivities on the $(\hat{x}, \hat{y})$-coordinates of the candidate points, that is, the real and imaginary parts of the $\hat{z}_j$. Thanks to complex differentiability, we know that changing $v_j \to v_j + \epsilon_\mathbb{C}$ and keeping $W$ fixed will change $\tilde{L}$ by $\epsilon_\mathbb{C} \cdot \sigma_{v_j}$ – and correspondingly for

$w_j$. Now, if we simultaneously change $(v_j, w_j) \to (v_j + \epsilon_\mathbb{R}, w_j + \epsilon_\mathbb{R})$ with $\epsilon_\mathbb{R}$ real, which corresponds to changing $(\hat{x}_j, \hat{y}_j) \to (\hat{x}_j + \epsilon_\mathbb{R}, \hat{y}_j)$, $\tilde{L}$ will change by $\epsilon_\mathbb{R} \cdot \sigma_{v_j} + \epsilon_\mathbb{R} \cdot \sigma_{w_j} = \epsilon_\mathbb{R} \cdot \sigma_{v_j} + \epsilon_\mathbb{R} \cdot \sigma_{v_j}^* = 2\epsilon_\mathbb{R} \mathrm{Re}\, \sigma_{v_j}$.

Likewise, changing $(\hat{x}_j, \hat{y}_j) \to (\hat{x}_j, \hat{y}_j + \epsilon_\mathbb{R})$ with $\epsilon_\mathbb{R}$ real changes $(v_j, w_j) \to (v_j + i\epsilon_\mathbb{R}, w_j - i\epsilon_\mathbb{R})$, and hence $\tilde{L}$ will change by $i\epsilon_\mathbb{R} \cdot \sigma_{v_j} - i\epsilon_\mathbb{R} \cdot \sigma_{w_j} = i\epsilon_\mathbb{R} \cdot \sigma_{v_j} - i\epsilon_\mathbb{R} \cdot \sigma_{v_j}^* = -2\epsilon_\mathbb{R} \mathrm{Im}\, \sigma_{v_j}$. This, then, tells us how to link up the real sensitivities (gradient components) with the complex sensitivities:

$$\sigma_{\hat{x}_j} = 2\mathrm{Re}\, \sigma_{v_j}, \quad \sigma_{\hat{y}_j} = -2\mathrm{Im}\, \sigma_{v_j}. \tag{21}$$

# B   DIFFERENCE OF MOMENTS

## B.1   MOMENTS AS SET FINGERPRINTS

Restricting our attention to one-dimensional points, an obvious choice for a permutation-invariant fingerprint would be moments of point sets. With $N = |Z|$, the $k$-th moment of $Z$ is $M_k(Z) := \sum_{i=1}^{N} z_i^k$. Knowing the values of these elementary-symmetric functions for $k = 1, \ldots, N$ completely determines the point set, making $m_Z = (M_1(Z), \ldots, M_N(z))^T \in \mathbb{R}^N$ a valid set fingerprint. Indeed, as explained in Appendix A of Zaheer et al. (2017), this is in some sense the *fundamental* permutation-invariant set fingerprint, since every symmetric function on sets of $N$ real numbers can be written in the form $\rho(m_Z)$ with some suitable function $\rho$. As explained there, this nicely parallels the Kolmogorov-Arnold theorem (Kolmogorov, 1956; Arnold, 1957) for the symmetric case.

However, this insight is only of limited use for constructing a practical permutation-invariant loss function, since numerical conditioning aspects have to be taken into account. Specifically, simply taking the $L_2$-distance between these moments fingerprints

$$L^{\mathrm{Mom}}(\hat{Z}, Z)^2 = \frac{1}{N} \sum_{k=1}^{N} \left[ \sum_{i=1}^{N} \hat{z}_i^k - \sum_{i=1}^{N} z_i^k \right]^2 \tag{22}$$

turns out to have fundamental problems matching up even moderately-sized real point sets, see Figure 3 for a $2 - d$ example. Beyond $\sim 7$ points, there are ways to collectively shift points such that the impact of the distortion on low-order moments is compensated, while the impact on high-order moments falls below the numerical accuracy threshold. Typically, the end result of a failed attempt to match up $N$ real points using $L^{\mathrm{Mom},2}$ will have points near zero being considerably off the target locations.

In the same experiment, the Holographic loss reliably matches up point sets with more than $40$ randomly sampled points. If we cast $\hat{Z} \mapsto L_U^{\mathrm{Hol}}(\hat{Z}, Z)^2$ into the $\rho(m_Z)$ language it takes on the form of a polynomial in all the point coordinates (in two dimensions: $\hat{z}_j = (\hat{x}_j, \hat{y}_j)$) of the general form

$$\rho(Z) = \sum_i c_i \hat{x}_1^{\xi_{i,1}} \hat{x}_2^{\xi_{i,2}} \cdots \hat{x}_N^{\xi_{i,N}} \cdots \hat{y}_1^{\eta_{i,1}} \hat{y}_1^{\eta_{i,2}} \cdots \hat{y}_N^{\eta_{i,N}} \tag{23}$$

with all exponents $\xi_{i,k}, \eta_{i,k}$ being either 0, 1, or 2. This largely avoids numerical problems due to the locations of zeroes of higher-degree polynomials being ill-conditioned w.r.t. coefficients. So, the "holographic" loss function can be considered as "maximally simple" in the sense of being a quadratic function when considering each input-coordinate separately. For large point sets, it may be useful to generalize the "holographic" loss function by also allowing weights for the contributions coming from different evaluation-points, in order to minimize the discrepancies between diagonal entries of the Hessian.

## B.2   MOMENTS-LOSS IN HIGHER DIMENSIONS

It is not obvious how one would extend the moments-based loss to higher dimensions. Using just the $N \cdot D$ moments for every coordinate, $\sum_{i=1}^{N} z_{i,j}^k$, will not give us a proper set fingerprint, since it does not discriminate between point sets obtained by shuffling the values of any one coordinate

between points. That is, it could not discriminate $\{(x_0, y_0); (x_1, y_1)\}$ from $\{(x_0, y_1); (x_1, y_0)\}$. A more appropriate choice are the moments

$$M_{(k_1, k_2, \ldots, k_D)}(Z) := \sum_{i=1}^{N} z_{i,1}^{k_1} z_{i,2}^{k_2} \cdots z_{i,D}^{k_D} \tag{24}$$

for $\sum_j k_j \leq N$. However, this scales infavorably with $N$ and one would naturally expect the same numerical problems we have seen in one dimension to persist in higher dimensions. This is confirmed in the investigations described next, where we used as a fingerprint the vector of all moments up to total scaling dimension $N$ in $D = 2$, i.e. $x^a y^b$ with $a + b \leq N$.

### B.3 Matching up Point Sets with Moments-Loss and Holographic Loss

In Figure 3 we show experiments matching up sets of randomly sampled points by minimizing $L^{\text{Mom},2}$ and $L^{\text{Hol},2}$ to numerical accuracy.

We see that, using 64-bit floating point numerics, the Holographic loss can in principle be used to match up sets with more than 40 randomly sampled points.

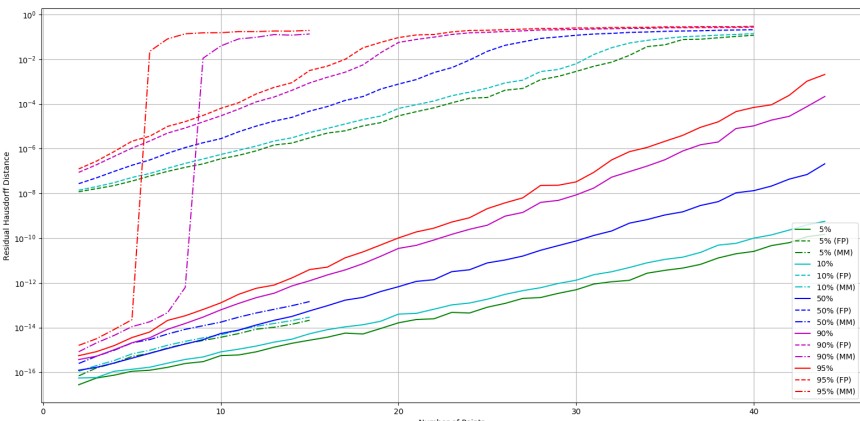

Figure 3: Performance of the 'holographic' loss function measured in terms of residual Hausdorff-distance between candidate- and target-set after minimizing $\log(10^{-100} + L_{U=Z}^{\text{Hol},2}(C, U))$ with SciPy's `scipy.optimize.fmin_bfgs` to `gtol` $10^{-8}$ for $1000$ examples (per number of points) of point-configurations $C, Z$ where each point is drawn uniformly from the 2-d unit disc. Solid lines: percentiles for minimizing as described. Dashed lines: percentiles when artificially limiting numerical accuracy of the potential and gradient by adding and then substracting again $1.0$. This effectively turns 64-bit floating point arithmetics into 53-bit fixed-point arithmetics by fixing the floating exponent. Under these conditions, the loss function behaves as is expected from condition number analysis, being useful to about $N = 15$ points. When allowing the floating point exponent to adapt, this loss function can in principle match up point sets well beyond $N = 40$ points when using 64-bit float arithmetics.

## C Miscellaneous

### C.1 Failure Case of One-Sided Hausdorff/SMD

Figure 4 depicts a simple failure case of the one-sided versions of the Hausdorff distance and the sum of minimal distances.

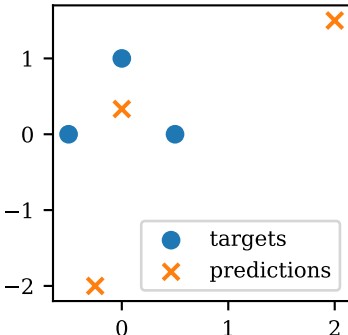

Figure 4: A configuration that leads to a local minimum of the target-centered onesided versions of the Hausdorff distance and the sum of minimal distances. All three target points have the same closest prediction point, which is located exactly in their "center of mass".

## C.2 EXAMPLE SOLUTIONS OF THE GRADIENT FLOW ODE

To gain some intuition for the dynamics induced by the Holographic loss function, we consider some example solutions of the gradient flow ODE

$$\hat{Z}(t=0) := \hat{Z}_{\text{start}}, \qquad \frac{d}{dt}\hat{z}_k(t) = -\left.\frac{\partial L^{\text{Hol}}(\hat{Z}, Z)^2}{\partial \hat{z}_k}\right|_{\hat{Z}(t)} \tag{25}$$

Figure 5 shows multiple instances of matching up sets of five two-dimensional points. Lines show how the prediction points move according to the ODE (25). Along each line, markers are placed indicating exponentially spaced time-steps $t_n = 2^n \cdot \Delta t$ with every fifth marker being drawn in black. (These do *not* correspond to iterations of gradient descent, but to the exact solution of the ODE. In the limit of infinitely small step size, the gradient descent dynamics would approach these curves.)

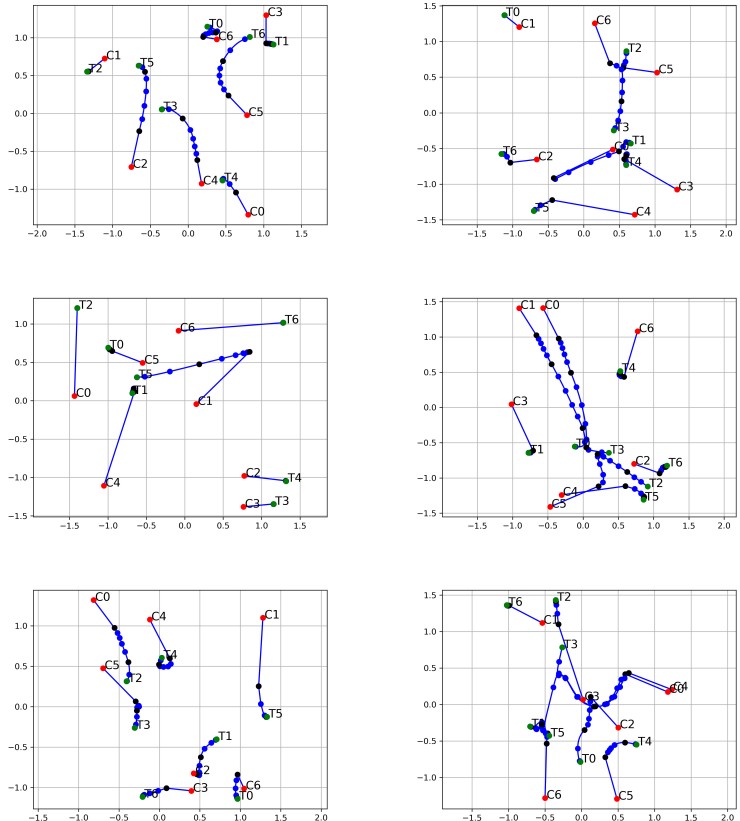

Figure 5: Gradient flows illustrating the behavior of the Holographic loss function. Left column: If there are reasonably clear ways to match up the two point sets, the gradient flow tends to find that identification and moves each point simultaneously in a meaningful way. This allows each entry of the gradient to forward meaningful information towards earlier layers in ML training, unlike with Hausdorff-distance. Right column: In more complex cases (such as two points almost coinciding), multiple points may travel alongside one another until the degeneracy finally gets broken.

