# OpenReview forum: "Holographic and other Point Set Distances for Machine Learning"
_ICLR.cc/2019/Conference_

### Official Review · AnonReviewer3 · 2018-11-03
**The paper is well written and is, in my opinion, a good contribution to the literature.**

**Rating:** 7
**Confidence:** 3

**Review:**

The authors introduce a novel distance function between point sets, based on the "permutation invariance" of the zeros of a polynomial, calling it "holographic" distance, as it essentially depends on all the points of the sets being compared. They also consider two other permutation invariant distances, and apply these in an end-to-end object detection task. These distance functions have time-complexity O(N^2) unlike the previously proposed "Hungarian distance" based on the Hungarian algorithm which is O(N^3) in general. Moreover, they authors show that in two dimensions all local minima of the holographic loss are global minima.

Pros: The paper is well written, the ideas are clearly and succinctly presented. Exploiting the connection between 2D point sets and zeros of polynomials is an interesting idea.

Cons: The experimental section could be better. For example, the authors could do simple experiments to show how an optimization algorithm would explore the holographic loss surface (in terms of hitting global/local minima) in dimensions greater than two. Also, in the object detection example, no comparison is given with the Hungarian loss based algorithm of Stewart et al. (2016) (at the very least, the authors could train their neural nets using the Hungarian loss, choosing one optimal permutation at the "transitioning points") .

---

### Official Review · AnonReviewer1 · 2018-11-05
**Insufficient**

**Rating:** 3
**Confidence:** 4

**Review:**

This paper proposes a new loss for points registration (aligning two point sets) with preferable permutation invariant property. For a 2D point set, the idea is to define a complex polynomial with the points (interpreted as complex numbers) as roots. To compare two point sets, the loss eventually boils down to evaluating the complex polynomial at all target points. The loss can be extended to high-dimensional point sets without the same theoretical framework. It is a sum over each target point the product of the distance of between the target point and all points of the source set. The claimed advantage of the proposed distance is it is more efficient compare to other metric such as Hausdorff distance or the sum of squared distance between matched point pairs (SMD). Experiments show that the proposed loss can be used in training.

The idea is interesting and new. However, the paper has a lot of room to improve. Here are a few specific issues that can be addressed in future version:

1) It is unclear what the advantage of the new idea is. I agree it leads to strong gradients compared to Hausdorff distance. However, this way of pulling all points to all points may also be very inefficient, as each point can get pulled toward all target points. These forces can easily cancel each other.

2) As already hinted in the paper, the gradient can be extremely unstable numerically. There is not any solution to address this issue.

3) Experimental results do not show the proposed method has any advantage over others. Also important loss such as the Wasserstein distance should be considered and compared with.

---

### Official Review · AnonReviewer2 · 2018-11-06
**Nice contribution but room for improvements**

**Rating:** 4
**Confidence:** 3

**Review:**

This paper proposes permutation invariant loss functions which depends on the distance of sets. They motivate this by problems where one wants to detect objects (say stones in an image) where there is no explicit ordering of objects.

Pros: The idea sounds interesting and seems like this approach could work well in practice. The authors provide an interesting algorithm which minimizes this loss function in O(N^2) instead of O(N^3). Moreover, the authors also show interesting properties of the Holographic loss, along with some interesting properties of the minima etc.

Cons: My major criticism of this work is that, while this seems like an interesting idea, the authors do not really provide extensive results on real world datasets. They consider a simple proof of concept which is based on MNIST images. I do not think this is sufficient. I would have been much more convinced with this paper if I had seen results on Pascal VOC and COCO, since this is the main motivation of their work.

---

### Meta-Review · Area_Chair1 · 2018-12-16
**Area chair recommendation**

**Confidence:** 5
**Recommendation:** Reject

**Metareview:**

This paper proposes permutation invariant loss functions which depend on the distance of sets. This has an interesting interpretation as the roots of a polynomial, and potentially leads to a more efficient method.

It is not clear, however, whether the method works well in practice for multiple reasons: (i) the experiments are performed in a limited setting, and the rebuttal specifically declined to consider more realistic datasets, (ii) there is an open question about the stability of the resulting gradients, which has been pointed out both in the paper and the reviews.


There was initially a majority vote for rejection. After author response, the only reviewer recommending acceptance wrote "As the other reviews (and my original review) say, the experimental results are not totally convincing. So I would not champion the paper in its present form."